# Fracture of Titanium Alloys at High Strain Rates and under Stress Triaxiality

**Vladimir V. Skripnyak \*, Evgeniya G. Skripnyak and Vladimir A. Skripnyak**

Department of Deformable Solid Body Mechanics, National Research Tomsk State University, 634050 Tomsk, Russia; eva.skrp@mail.ru (E.G.S.); skrp2006@yandex.ru (V.A.S.)
\* Correspondence: skrp2012@yandex.ru; Tel.: +7-923-4027744

**Abstract:** The present study investigates the effect of stress triaxiality on mechanical behavior and fracture of Ti-5Al-2.5Sn alloy in a practical relevant strain rate range from 0.1 to 1000 s$^{-1}$. Tensile tests were carried out on flat smoothed and notched specimens using an Instron VHS 40/50-20 servo-hydraulic test machine. High-speed video registration was conducted by Phantom 711 Camera. Strain fields on the specimen gauge area were investigated by the digital image correlation method (DIC). The fracture surface relief was studied using digital microscope Keyence VHX-600D. Stress and strain fields during testing of the Ti-5Al-2.5Sn alloy were analyzed by the numerical simulation method. The evolution of strain fields at the investigated loading condition indicates that large plastic deformation occurs in localization bands. The alloy undergoes fracture governing by damage nucleation, growth, and coalescence in the localized plastic strain bands oriented along the maximum shear stresses. Results confirm that the fracture of near alpha titanium alloys has ductile behavior at strain rates from 0.1 to 1000 s$^{-1}$, stress triaxiality parameter $0.33 < \eta < 0.6$, and temperature close to 295 K.

**Keywords:** stress triaxiality; notched specimen; damage evolution; titanium alloys; high strain rate

---

## 1. Introduction

Polycrystalline alpha-titanium alloys belong to the isomechanical group of metal materials with hexagonal close packed (HCP) crystalline lattice. Titanium alloys are widely employed in high-end industries like aerospace engineering, power engineering, and medicine because of the combination of high specific strength and good corrosion resistance. Materials belonging to the same mechanical group exhibit similar mechanical behavior in wide ranges of strain rates and temperatures owing to the similarity in mechanisms of plastic deformation and fracture [1–3].

Alpha-titanium alloys exhibit enhanced high-temperature creep resistance, minor heat treatment hardening, and lower ductility as compared with α+β and β titanium alloys [2].

Among various developed titanium alloys, the Ti-5Al-2.5Sn alloy (Grade 6 or analogue VT5-1 alloy) is especially famous for its high fracture toughness at cryogenic environments, excellent weldability, and outstanding creep resistance at elevated temperatures.

The interest in the manufacture of Ti-5Al-2.5Sn alloy parts by selective laser melting (SLM) technology has increased significantly recently, not only because of the ability to fabricate products of high quality but also because of its potential to reduce time-to-market and increase procurement volume [4].

Generalization of data on laws of deformation and fracture of alpha titanium alloys in a wide range of strain rates are useful for the engineering analysis of titanium structural elements under dynamic impacts [5–8]. The ductility and strength of titanium alloys in a wide range of strain rates depend on the grain size and distribution [9–12]. There is evidence that the ductility and fracture of the

hexagonal close-packed polycrystalline metals and alloys are strongly dependent on the accumulated plastic strain and stress triaxiality [13–17].

A significant distinction has been noted between the regimes of high and low stress triaxiality. High values of triaxiality (i.e., $\eta > 1.5$) may be achieved in local areas, such as at the ends of the cracks or in the center of the neck or notch under tension. Low stress triaxiality takes place at surfaces and protruding corners, where the equivalent shear stress is high relative to the hydrostatic pressure [18–22]. Fracture initiation is strongly inhibited at low stress triaxiality.

Several models were proposed to investigate the effect of triaxiality on the fracture of polycrystalline metals and alloys [23–27]. Neilsen and Tvergaard [23,24] showed that ductile fracture can be described by means of the criterion incorporating the stress triaxiality and the Lode angle. Valoppi [14] used the phenomenological Johnson–Cook hardening model and damage initiation criterion with an energy-based law describing the damage evolution of titanium alloys. In this research, we study the influence of different values of stress triaxiality ($0.33 < \eta < 0.6$) on the ductile fracture in a practical relevant range of strain rates from 0.1 to 1000 $s^{-1}$.

## 2. Experimental Procedure

### 2.1. Quasistatic and Dynamic Tensile Tests

The Ti-5Al-2.5Sn alpha titanium alloy was studied under tension in a range of strain rates from 0.1 to 1000 $s^{-1}$ and at room temperature. Specimens were cut out from a sheet of titanium alloy by means of electro-erosion method. The thickness of the sheet sample was $1.3 \pm 0.05$ mm. Specimens of Ti-5Al-2.5Sn alloys had an average grain size of 40 μm. The initial gauge length $l_0$ was equal to $20 \pm 0.1$ mm.

Figure 1 shows geometry of specimens. The minimal cross-sectional area of flat specimens ($w \times d$) was $A_0 = 7.8 \pm 0.05$ mm$^2$, and notched flat specimens had notch radii of $R = 10$, 5, and 2.5 mm, respectively.

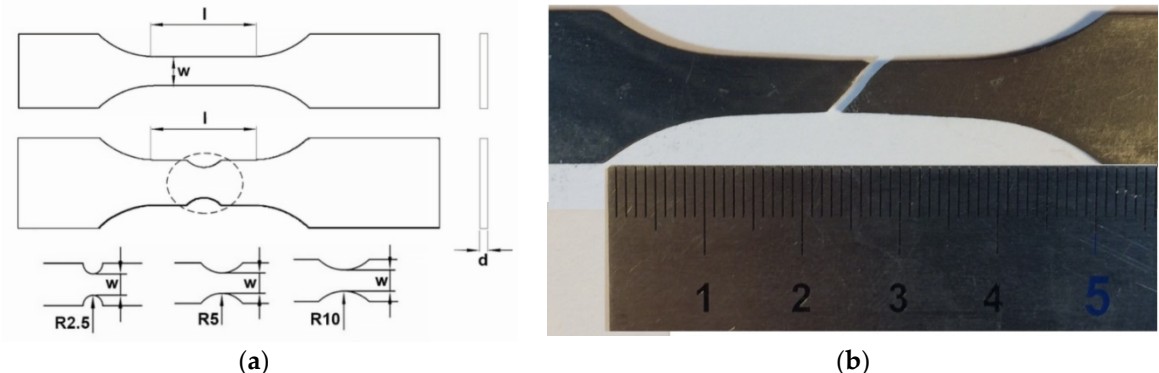

(**a**)          (**b**)

**Figure 1.** Schematic of notched specimen. (**a**) Initial geometry of specimen and (**b**) photo of the fractured sample.

The tension tests were carried out in a range of strain rates (0.1–1000 $s^{-1}$) at room temperature using the Instron high speed testing machine VHS 40/50-20 (Instron, High Wycombe, UK) with a 50 kN load cell.

Tests were conducted in the velocity controlled mode at crosshead speed of $0.002 \pm 0.0002$, $2 \pm 0.01$, and $20 \pm 0.1$ m/s, respectively.

The tensile force and displacement were recorded at high temporal resolution up to complete fracture of the specimen.

True stress was determined by analytical relations [16]:

$$\sigma_1^{true} = (F/A_0)(1 + \Delta l/l_0) \tag{1}$$

where $\sigma_1^{true}$ is the true stress, $F$ is the tensile force, $A_0$ is initial minimal cross sectional area of the specimen, $\Delta l$ and $l_0$ are the elongation and the initial length of the specimen gauge part.

True strain over the gauge length was determined by relationship [16]:

$$\varepsilon_1^{true} = \ln(1 + \Delta l/l_0) \tag{2}$$

where $\varepsilon_1^{true}$ is true strain.

The stress triaxiality $\eta$ is defined as [13]:

$$\eta = p/\sigma_{eq}, \tag{3}$$

where $p = \sigma_I/3$ is the pressure, $\sigma_I = \sigma_{11} + \sigma_{22} + \sigma_{33}$ and $\sigma_{II} = \sigma_{11}\sigma_{22} + \sigma_{22}\sigma_{33} + \sigma_{11}\sigma_{33} - \sigma_{12}^2 - \sigma_{23}^2 - \sigma_{13}^2$ are first and second invariants of the Cauchy stress tensor, respectively, and $\sigma_{eq} = (3\sigma_{II})^{1/2}$ is the equivalent stress.

The initial value of the stress triaxiality $\eta$ was calculated by the analytical formula in the plane stress state [17]:

$$\eta = (1 + 2A)/(3\sqrt{A^2 + A + 1}, \ A = \ln[1 + w/(4R)] \tag{4}$$

where w is the minimal width of the specimen at the notched area and $R$ is the notch radius.

The initial value of the stress triaxiality $\eta$ was varied in the range 0.33–0.6.

The stress triaxiality $\eta$ will be different from the initial value when the sample neck is formed.

Fields of deformation of Ti-5Al-2.5Sn specimens were obtained by means of the digital image correlation (DIC) method. In order to measure fields of displacement by DIC method, paint patterns were applied to the specimen surface, which enabled us to compare the surface texture in frames of high-speed video registration.

The high-speed Phantom V711 camera (Vision Research - AMETEK Co., Wayne, NJ, USA) was used to record the specimen deformation at a speed of 100 thousand frames per second.

The application of the DIC method has enabled us to determine the deformation field of notched specimens and directly observe the influence of the notch radius on the strain distribution.

Three tests were carried out for each specimen type at strain rates of 100 s$^{-1}$ and 0.1 s$^{-1}$, and five tests were carried out at strain rates of 1000 s$^{-1}$. In each test series, a high degree of reproducibility of the recorded loading rate, force, and displacement was observed.

*2.2. Numerical Simulation of Plastic flow and Damage Evolution*

The computational model uses the theoretical basis of continuum damage mechanics.

The system of Equations includes:

Conservation Equations (5),
Kinematic relations (6),
Constitutive relations (7),
Equation of State (8),
Relaxation Equation for the deviatoric stress tensor (9).

$$\frac{d\rho}{dt} = \rho\frac{\partial u_i}{\partial x_i}, \ \frac{\partial \sigma_{ij}}{\partial x_j} = \rho\frac{du_i}{dt}, \ \rho\frac{dE}{dt} = \sigma_{ij}\dot{\varepsilon}_{ij} \tag{5}$$

$$\dot{\varepsilon}_{ij} = (1/2)[\partial u_i/\partial x_j + \partial u_j/\partial x_i), \ \dot{\omega}_{ij} = (1/2)[\partial u_i/\partial x_j - \partial u_j/\partial x_i) \tag{6}$$

$$\sigma_{ij} = \sigma_{ij}^{(m)}\varphi(f), \ \sigma_{ij}^{(m)} = -p^{(m)}\delta_{ij} + S_{ij}^{(m)} \tag{7}$$

$$P^{(m)} = P_x^{(m)}(\rho) + \Gamma(\rho)\rho E_T , \; E_T = C_P T ,$$
$$P_x^{(m)} = \tfrac{3}{2}B_0 \cdot ((\rho_0/\rho)^{-7/3} - (\rho_0/\rho)^{-5/3})[1 - \tfrac{3}{4}(4 - B_1) \cdot ((\rho_0/\rho)^{-2/3} - 1)]$$

(8)

$$DS_{ij}^{(m)}/Dt = 2\mu(\dot{\varepsilon}_{ij}^e - \delta_{ij}\dot{\varepsilon}_{kk}^e/3) ,$$
$$\dot{\varepsilon}_{ij} = \dot{\varepsilon}_{ij}^e + \dot{\varepsilon}_{ij}^P , \; \dot{\varepsilon}_{ij}^P = \dot{e}_{ij}^P + \delta_{ij}\dot{\varepsilon}_{kk}^P/3, \; \dot{e}_{ij}^P = \lambda \partial\Phi/\partial\sigma_{ij}, \; \dot{\varepsilon}_{kk}^P = \dot{f}_{growth}/(1 - f)$$

(9)

where $\rho$ is the mass density, $u_i$ is the components of the particle velocity vector, $x_i$ is Cartesian coordinates, $I = 1, 2, 3$, $E$ is the specific internal energy, $\dot{\varepsilon}_{ij}$ , $\dot{\omega}_{ij}$ are the components of strain rate tensor and the bending–torsion tensor, the $\varphi(f)$ function establishes a relation between the effective stresses of the damaged medium and the stresses in the condensed phase, $\Gamma$ is the Grüneisen coefficient, $\rho_0$ is the initial mass density of the condensed phase of the alloy, $\gamma_R, \rho_R, n, B_0, B_1$ are the material's constants, $C_p$ is the specific heat capacity, $D(\cdot)/Dt$ is the Jaumann derivative, $\mu$ is the shear modulus, $\dot{f}_{growth}$ is the void growth rate, $f$ is the void volume fraction in damaged medium, $\dot{\lambda}$ is the plastic multiplier derived from the consistency condition $\dot{\Phi} = 0$, and $\Phi$ is the plastic potential. The plastic potential was described using the Gurson–Tvergaard model (GTN) [13,23,24].

The function $\varphi(f)$ takes the form of $(1 - f)$ for pressure and is implicitly defined for the deviatoric stress tensor [23,24].

The temperature rise associated with energy dissipation during plastic flow can be evaluated by relation [28]:

$$T = T_0 + \int_0^{\varepsilon_{eq}^P} (\beta/\rho\,C_p)\,\sigma_{eq}d\varepsilon_{eq}^P$$

(10)

where $T_0$ is the initial temperature and $\beta \sim 0.9$ is the parameter representing a fraction of plastic work converted into heat.

The specific heat capacity for Ti-5Al-2.5Sn titanium was calculated by the phenomenological relations within the temperature range 293–1115 K [29–32].

$$C_p = 248.389 + 1.53067T - 0.00245T^2 \; [\text{J/kgK}] \; \text{at} \, 0 < T < T_{\alpha\beta} = 1320 \, \text{K}$$

(11)

The temperature dependence of the shear modulus for alpha titanium alloy was described by the equation:

$$\mu(T) = 48.66 - 0.03223\,T\,[GPa] , \; (273\,\text{K} < T < 1200\,\text{K}).$$

(12)

The flow stress was described by equation:

$$\sigma_s = \sigma_{s0}\exp\{C_1\sqrt{(1 - T/T_m)}\} + C_2\sqrt{1 - \exp\{-k_0\varepsilon_{eq}^P\}}\exp\{-C_3T\}\exp\{C_4T\ln(\dot{\varepsilon}_{eq}/\dot{\varepsilon}_{eq0})\}$$

(13)

where $\dot{\varepsilon}_{eq} = [(2/3)\dot{\varepsilon}_{ij}\dot{\varepsilon}_{ij}]^{1/2}$, $\dot{\varepsilon}_{eq0} = \gamma_1\exp\{-T/\gamma_2\} + \gamma_3$, $\varepsilon_{eq}^P = \int_0^t \dot{\varepsilon}_{eq}^P dt$ is the plastic strain intensity $\gamma_1 = 2115.08615 \, \text{s}^{-1}$, $\gamma_2 = 38.26589 \, \text{K}$, $\gamma_3 = 9.82388 \, 10^{-5} \, \text{s}^{-1}$ and $T_m$ is the melting temperature.

Material coefficients of Ti-5Al-2.5Sn are shown in Table 1.

**Table 1.** Material coefficients.

| Coefficients | $\sigma_{s0}$, GPa | $C_1$ | $C_2$, GPa | $C_3$, K$^{-1}$ | $C_4$, K$^{-1}$ | $k_0$ | $T_m$, K |
|---|---|---|---|---|---|---|---|
| Ti-5Al-2.5Sn (Grade 6) | 0.02 | 3.85 | 0.56 | 0.0016 | 0.00009 | 8.5 | 1875 |

### 2.3. Damage Model

　　Although deformation fields can be determined using the DIC method, direct observation of the origin and development of damage at the mesoscopic level is a technical difficulty. Therefore, numerical simulation analysis is required to obtain information on the evolution of damage during the plastic deformation. In this paper, numerical modeling was carried out to quantitatively identify the influence of the triaxial stress state on the development of damage and ductile fracture of a titanium alloy under tension and high-triaxiality tension loading conditions.

　　The influence of damage on the flow stress was taken into account using the Gurson–Tvergaard model [13,23,24]:

$$(\sigma_{eq}{}^2 / \sigma_s{}^2) + 2q_1 f^* \cosh(-q_2\, p/2\sigma_s) - 1 - q_3(f^*)^2 = 0 \tag{14}$$

where $\sigma_s$ is the yield stress and $q_1$, $q_2$, and $q_3$ are model parameters.

　　The Needleman model was used to describe the damage kinetics [19].

　　The rate of void growth is obtained by assuming mass conservation and depends on the volume change part of the plastic strain. Consequently, there is no void growth in pure shear deformation. The void nucleation depends on the equivalent plastic strain $\varepsilon^{\mathrm{P}}{}_{eq}$ [33].

$$\begin{aligned}
\dot{f} &= \dot{f}_{nucl} + \dot{f}_{growth}, \\
\dot{f}_{nucl} &= \varepsilon^{\mathrm{P}}{}_{eq}(f_N/s_N)\, \exp\{-0.5\,[\varepsilon^{\mathrm{P}}{}_{eq} - \varepsilon_N)/s_N]^2\}, \\
\dot{f}_{growth} &= (1-f)\dot{\varepsilon}^{\mathrm{P}}_{kk},
\end{aligned} \tag{15}$$

where $\varepsilon_N$ and $s_N$ are the average nucleation strain and the standard deviation, respectively. The amount of nucleating voids is controlled by the parameter $f_N$.

$$\begin{aligned}
f^* &= f \text{ if } f \le f_c; \\
f^* &= f_c + (\overline{f}_F - f_c)/(f_F - f_c) \text{ if } f > f_c,
\end{aligned} \tag{16}$$

where $\overline{f}_F = (q_1 + \sqrt{q_1{}^2 - q_3})/q_3$.

　　The final stage in ductile fracture comprises the voids coalescence. This causes softening of the material and accelerated growth rate of the void fraction $f^*$.

　　The model parameters for Ti-5Al-2.5Sn titanium alloy were determined via numerical simulation. Numerical values of model parameters were fitted by combined experimental and numerical techniques [33].

　　Numerical values of model parameters are given in Table 2.

**Table 2.** Dimensionless parameters for the Gurson–Tvergaard (GTN) model for alpha titanium alloys.

| Parameters Equations (8) and (9) | $q_1$ | $q_2$ | $q_3$ | $f_0$ | $f_N$ | $f_c$ | $f_F$ | $\varepsilon_N$ | $s_N$ |
|---|---|---|---|---|---|---|---|---|---|
| Ti-5Al-2.5Sn (Grade 6) | 1.3 | 1 | 1.69 | 0.00 | 0.2 | 0.035 | 0.4 | 0.28 | 0.1 |

　　Computational domains were meshed with eight-node linear bricks and reduced integration together with hourglass control.

　　Because mesh densities may affect the damage process, fine mesh with the edge length of 0.06 mm was applied at the central local zone of notched specimens, and the edge length of 0.5 mm was applied in other zones. Scheme of boundary conditions is shown in Figure 2.

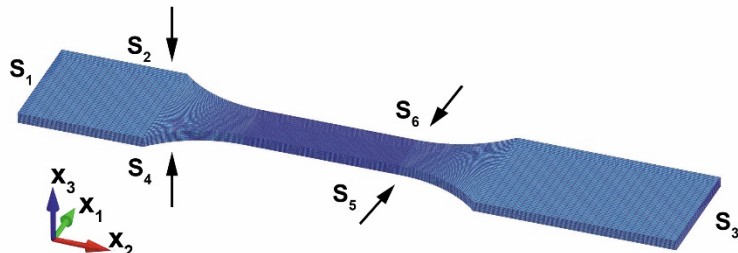

**Figure 2.** Scheme of boundary conditions.

Boundary conditions corresponding to uniaxial tension of the specimen at a constant strain rate have the form:

$$u_1|_{S_1} = 0, \quad u_1|_{S_3} = 0,$$
$$u_2|_{S_1} = 0, \quad u_2|_{S_3} = v_2,$$
$$u_3|_{S_1} = 0, \quad u_3|_{S_3} = 0,$$
$$\sigma_{ij}\big|_{S_2 \cup S_4 \cup S_5 \cup S_6} = 0,$$

(17)

where $u_i|_{S_j}$ is the components of the particle velocity vector on the surface $S_j$ and $v_2$ is the tensile velocity.

The initial conditions correspond to the free stress state of the material in a uniform temperature field.

The computer simulations were performed with the use of LS DYNA (ANSYS WB 15.2, ANSYS, Inc., Canonsburg, PA, USA) software. The calculations were carried out using the finite-difference scheme of second order accuracy.

## 3. Results

### 3.1. Results of Tensile tests

Figure 3 shows experimental force–elongation diagrams of specimens.

Figure 3a shows force–displacement diagrams obtained for smooth specimens. The yield stress of Ti-5Al-2.5Sn increases, but the ultimate elongations do not change significantly under tension within the range of strain rate from 0.1 to 1000 s$^{-1}$.

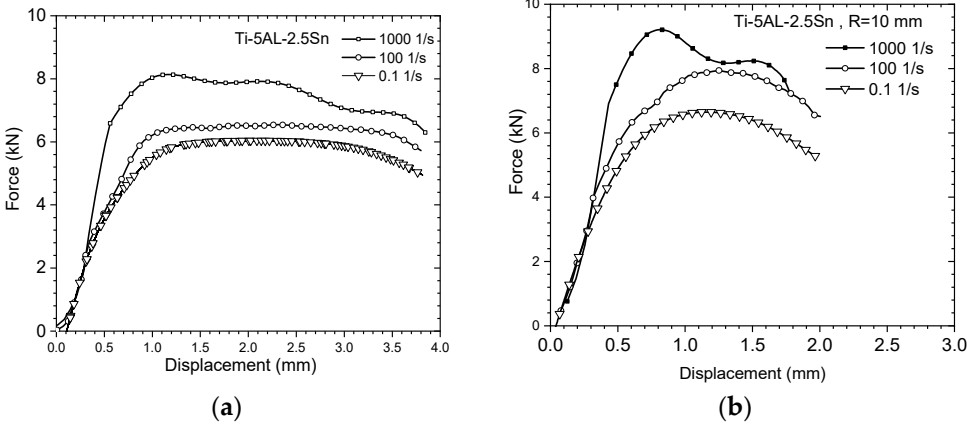

**(a)**　　　　　　　　　　　　　　　**(b)**

**Figure 3.** *Cont.*

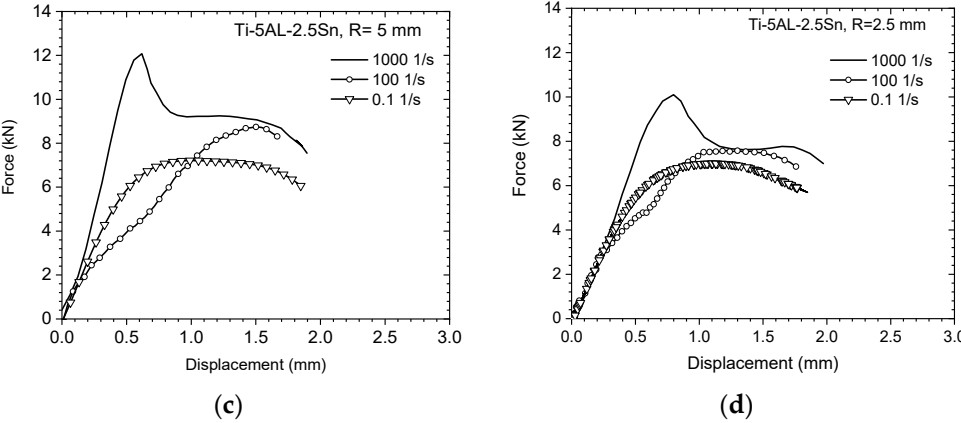

**Figure 3.** Force versus displacement of Ti-5Al-2.5Sn specimens under tension at strain rates of 0.1, 100, and 1000 s$^{-1}$; (**a**) specimens without notch; (**b**) specimens with notch radius $R = 10$ mm; (**c**) $R = 5$ mm; and (**d**) $R = 2.5$ mm.

The elongation to fracture of the alloy under tension at strain rates from 0.1 to 100 s$^{-1}$ decreases with increasing initial value of the stress triaxiality $\eta$. Note that the indicated decrease in elongation to fracture has no monotonic behavior with increasing strain rate. Results of tests have shown increased elongation to fracture in the gauge zone of specimens at an average strain rate of 1000 s$^{-1}$ as compared with quasi-static loading conditions. Increasing strain to fracture with increasing strain rate may be caused by the intensification of twinning mechanisms governing plastic deformation. This phenomenon takes place in a rather limited range of strain rates and temperatures, as higher strain rates will result in the necessity for additional energy dissipation channels, such as damage nucleation. Delay in the growth of damage in the plastic shear bands may be caused by local temperature raise.

The increase in temperature within the localized shear bands in Grade 1 titanium alloys was earlier observed by infrared thermometry measurement [11]. Note that the presence of a falling branch on the force–elongation diagrams was repeatedly observed in experiments at a strain rate of 1000 s$^{-1}$ in notched specimens. The indicated deformation behavior shows reproducibility achieved by testing a series of specimens and reflect the specific phenomenon of high-speed deformation of titanium alloys in the presence of stress concentrators. Figure 4 shows true stress–true strain diagrams obtained from data on forces–displacement and fields of deformation measured by the DIC method.

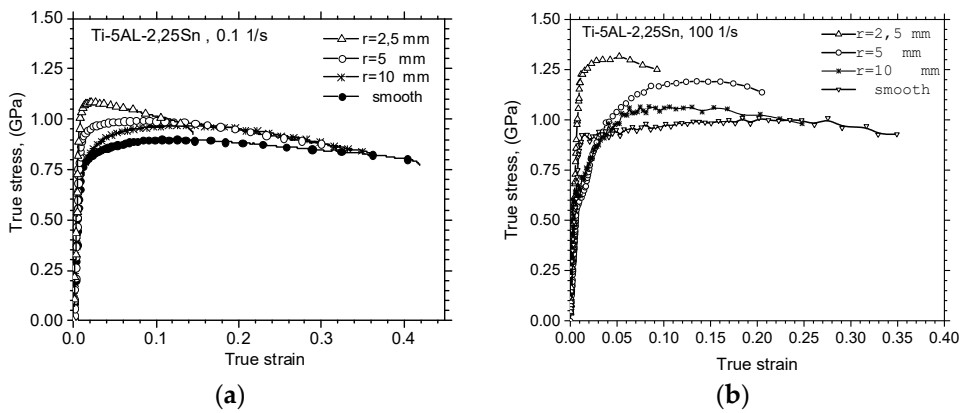

**Figure 4.** *Cont.*

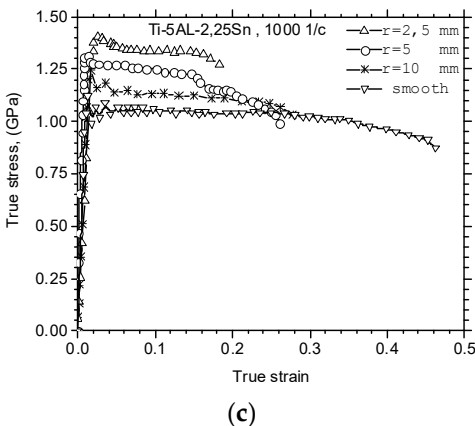

(c)

**Figure 4.** True stress versus true strain of Ti-5Al-2.5Sn specimens with notch radii of 10, 5, 2.5 mm at strain rates (**a**) 0.1 s$^{-1}$, (**b**) 100 s$^{-1}$, and (**c**) 1000 s$^{-1}$.

Results on quasi-static and high-speed deformation of Ti-5Al-2.5Sn alloy are in good agreement with recent results of Zhang et al. [31,32] on uniaxial tension and dynamic compression. Results of high triaxiality tension show increasing plastic flow stress with decreasing notch radius at strain rates of 0.1, 100, and 1000 s$^{-1}$. This indicates a significant strain hardening of near alpha titanium alloys observed under quasi-static and at high strain rate loading conditions [28,29,31,32].

It should be noted, that force–displacement diagrams obtained by standard methodologies are used to obtain data on stresses and strains averaged over the gauge length. In tensile tests of notched samples, fields of strain and stress are inherently heterogeneous, thus averaged values do not reflect the features of mechanical behavior of alloys during deformation. This circumstance is important for understanding the laws of damage evolution and fracture of alloys in accordance with variation of the stress state triaxiality. Taking this circumstance into account is important for obtaining realistic engineering analysis projections of parts fabricated from titanium alloys.

### 3.2. Strain Fields Observation by Digital Image Correlation (DIC)

Frames of high-speed video registration were used to analyze the deformation of Ti-5Al-2.5Sn under high speed tension of smooth and notched specimens. Videos were recorded at several resolutions: 1280 × 800, 1024 × 680, and 512 × 400 pixels at strain rates of 0.1, 100, and 1000 s$^{-1}$, respectively. Image size was varied with the resolution to record images with 250 pixels along minimal width of the specimen gauge part. In all DIC analyses performed, the subset size was set to 12 pixels, which resulted in noise-free strain fields results. The use of gray-scale produces an additional smooth effect on artificial noises and helps to emphasize the plastic flow behavior. Figures 5–8 show the equivalent strain fields measured by the DIC method on the surface of smooth specimens and specimens with the notch radius, *R* = 10, 5, and 2.5 mm, respectively.

In Figures 5–8, subfigures (a)–(c) correspond to strain rates of 0.1, 100, and 1000, respectively. In order to show the features of the development of plastic deformation, these subfigures are divided into subsections (I)–(III). The subsection number increases with the deformation time.

The first subsection indicates strain field heterogeneity to form in the work area. The second subsection shows the formation of conjugate localized shear bands. The third subsection shows the strain localization zone in which the fracture crack is formed. The application of high-speed video registration enabled us to record processes of crack formation in zones near stress concentrations.

An analysis of the equivalent strain fields indicates decreasing ultimate strain values within the localized region with increasing stress triaxiality parameter *η*. The orientation of plastic strain localization bands changes in the vicinity of the stress concentration zone. Localization bands are oriented at an angle close to 45° to the tension axis. Fracture of the sample occurs as a result of crack formation in the localized plastic strain area. The orientation of strain localization bands and

forming crack in the notched specimens changes when the stress triaxiality parameter $\eta$ changes. In a specimen with the notch radius $R = 2.5$ mm, the orientation of conjugate shear bands and forming crack approaches 90° to the tension axis. An analysis of the equivalent strain field patterns has shown that the crack formation in alpha titanium alloys is preceded by the formation of localized plastic strain bands. Obtained results on high-speed deformation testing indicate that the fracture character remains ductile even in the presence of stress concentrators.

Comparison of strain fields at 100 and at 1000 s$^{-1}$ indicates increasing ultimate equivalent plastic strain in localized shear bands with increasing strain rates.

An analysis of the equivalent strain fields indicates that maximum values are always achieved in the zone of intersection of localization bands.

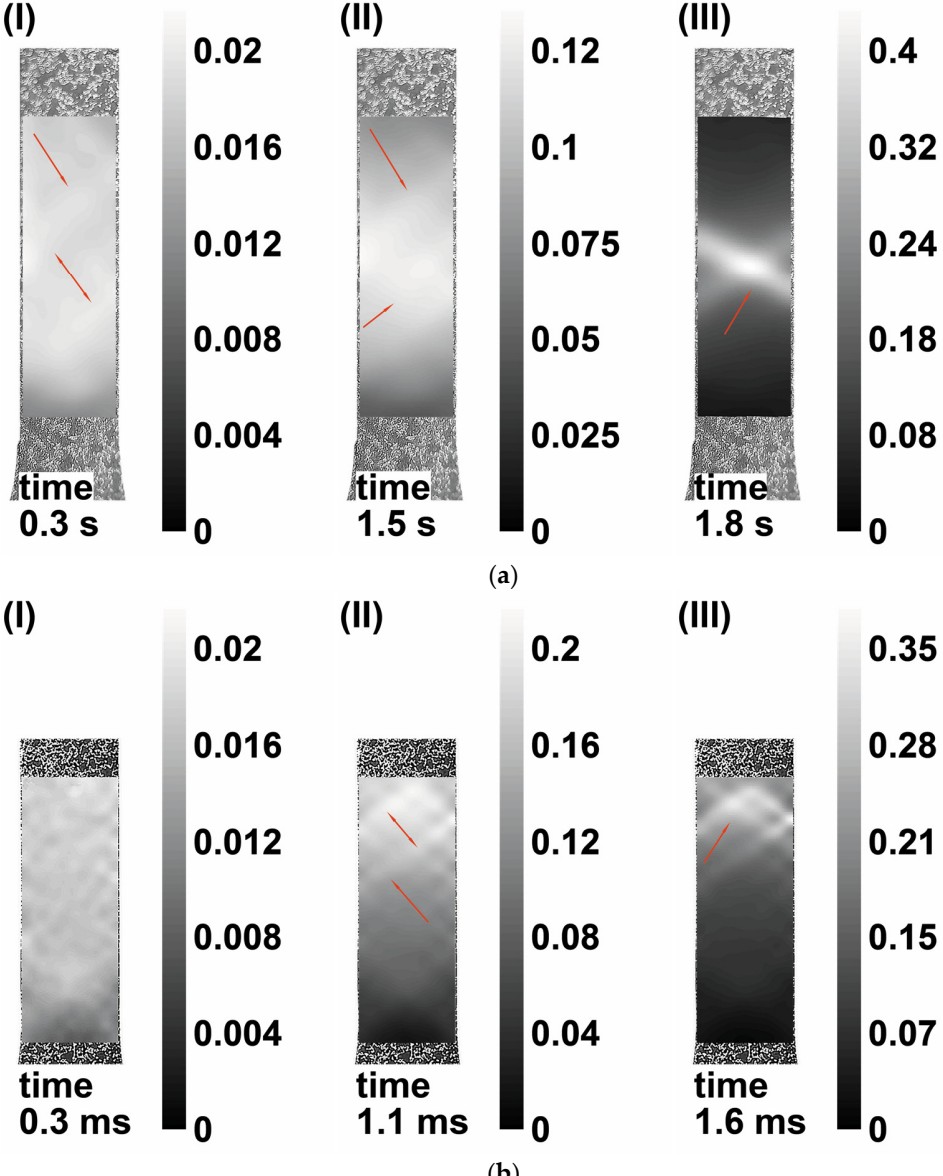

**Figure 5.** *Cont.*

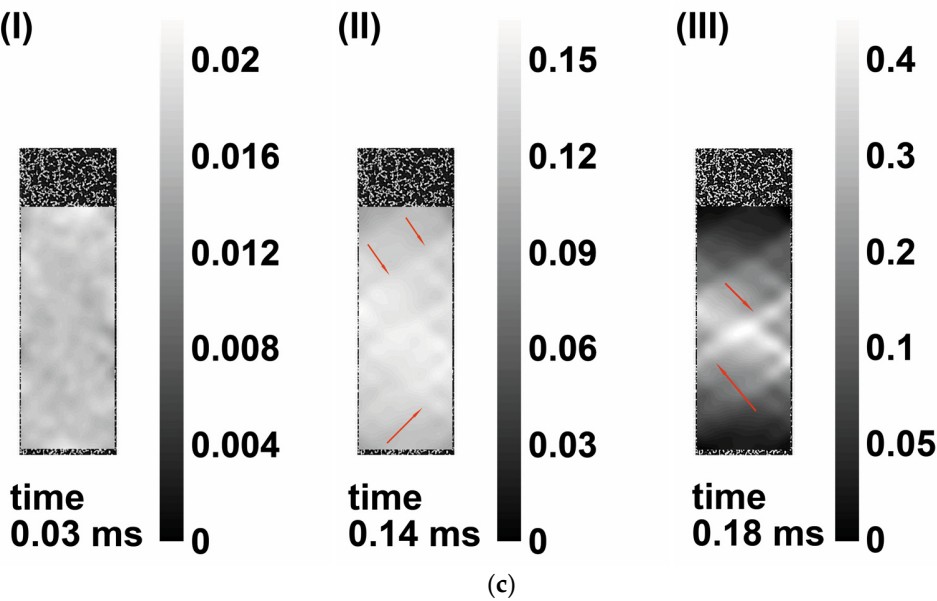

(**c**)

**Figure 5.** Fields of equivalent strain of smooth specimens at (**a**) 0.1 s$^{-1}$, (**b**) 100 s$^{-1}$, and (**c**) 1000 s$^{-1}$.

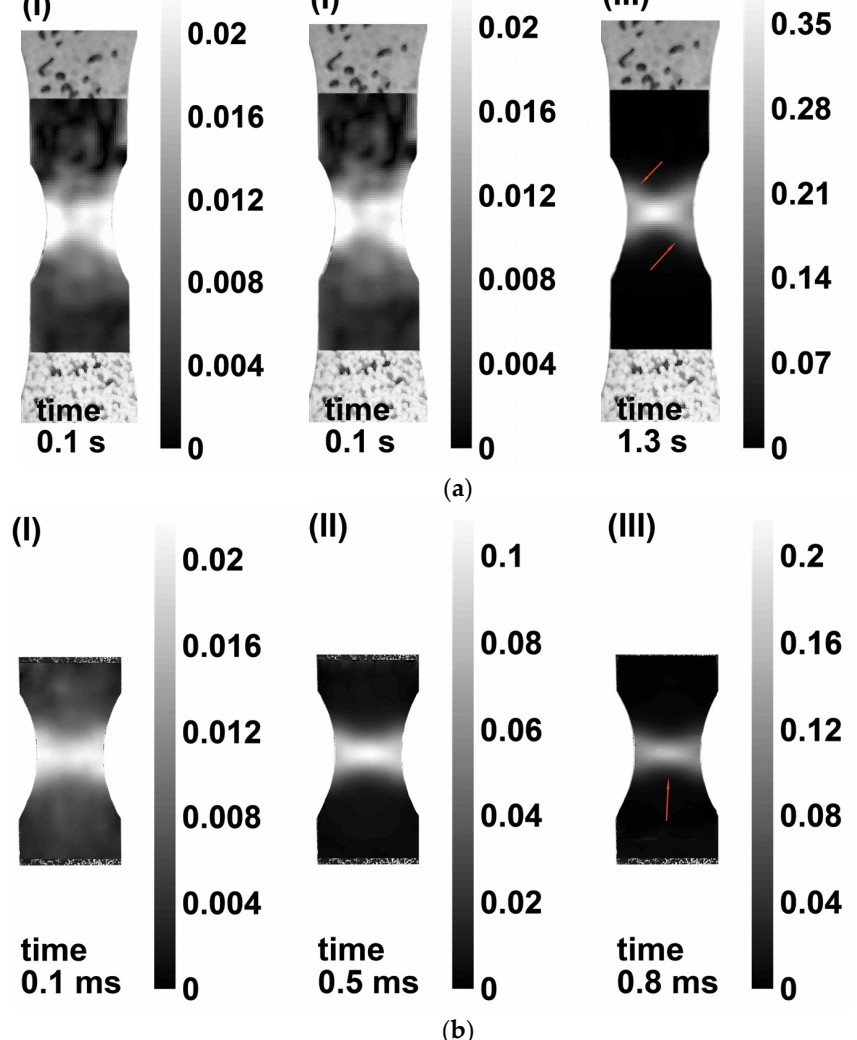

**Figure 6.** *Cont.*

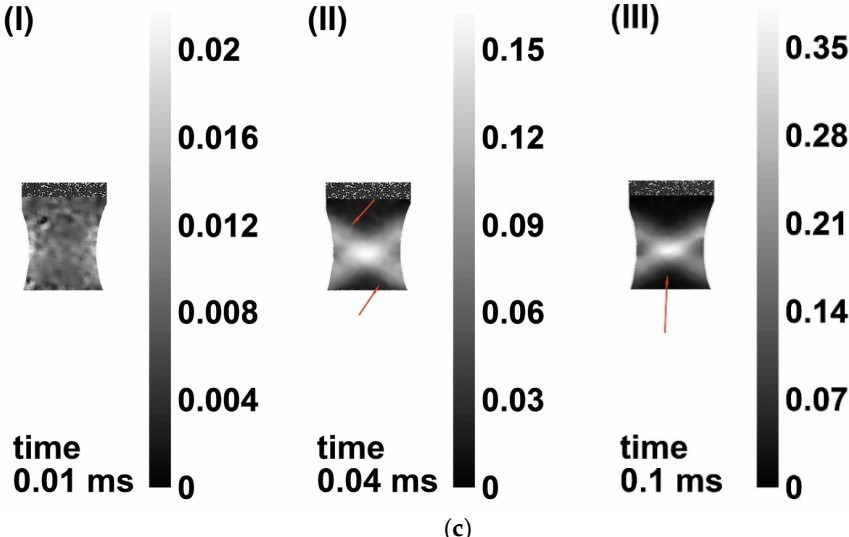

(**c**)

**Figure 6.** Fields of equivalent strain of specimens with the notch radius of 10 mm at (**a**) 0.1 s$^{-1}$, (**b**) 100 s$^{-1}$, and (**c**) 1000 s$^{-1}$.

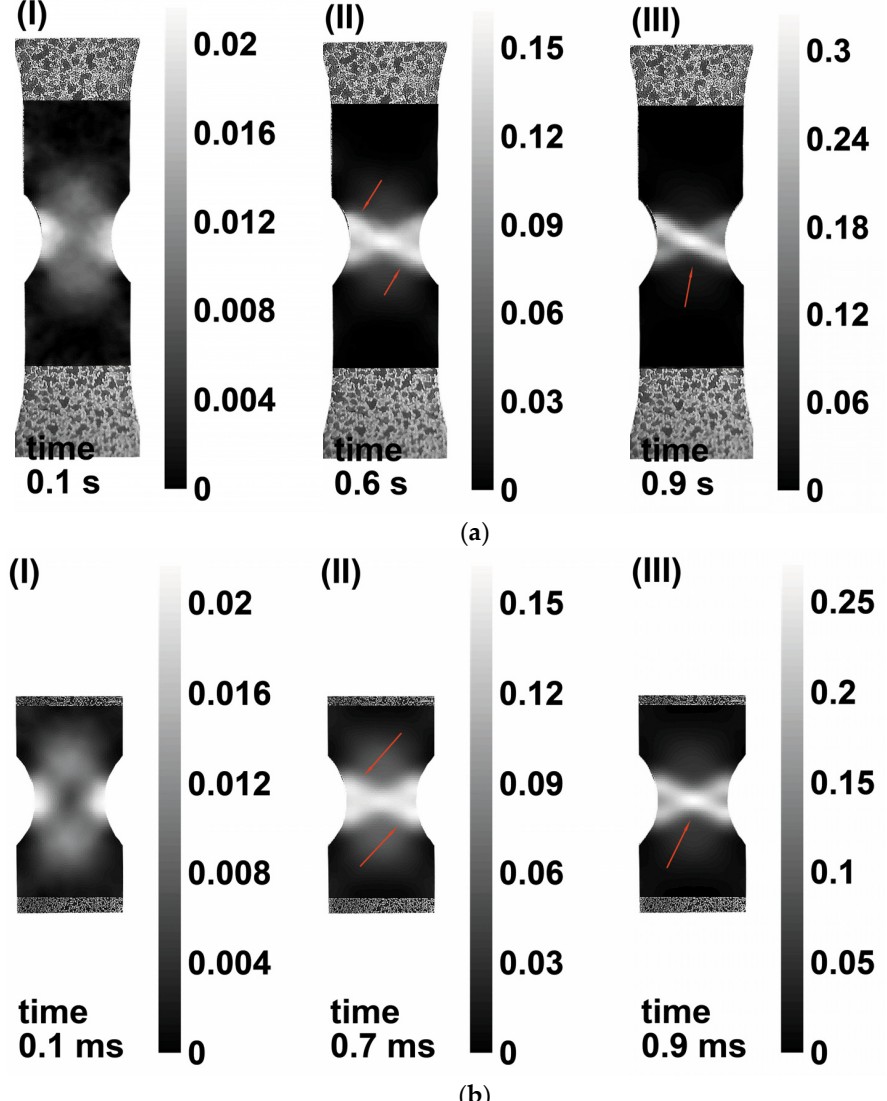

(**a**)

(**b**)

**Figure 7.** *Cont.*

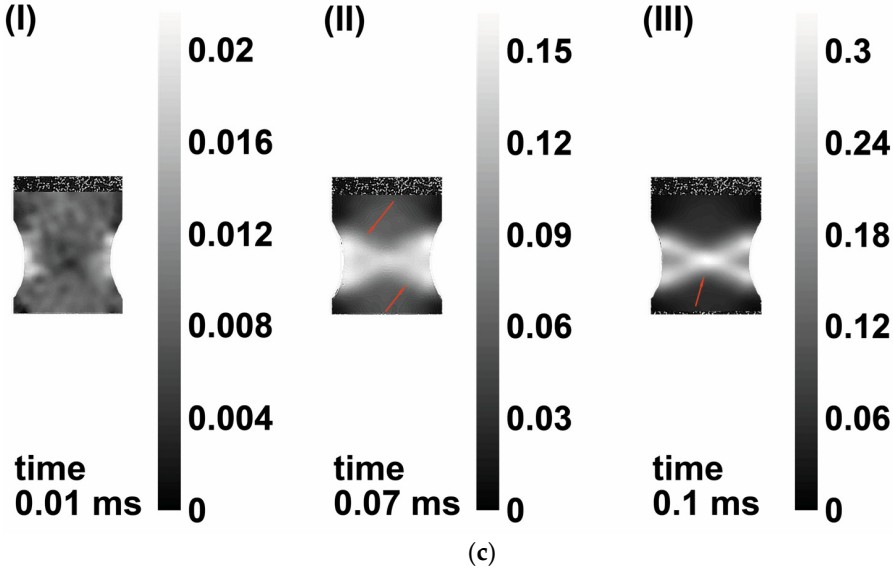

(c)

**Figure 7.** Fields of equivalent strain of specimens with the notch radius of 5 mm at (**a**) 0.1 s$^{-1}$, (**b**) 100 s$^{-1}$, and (**c**) 1000 s$^{-1}$.

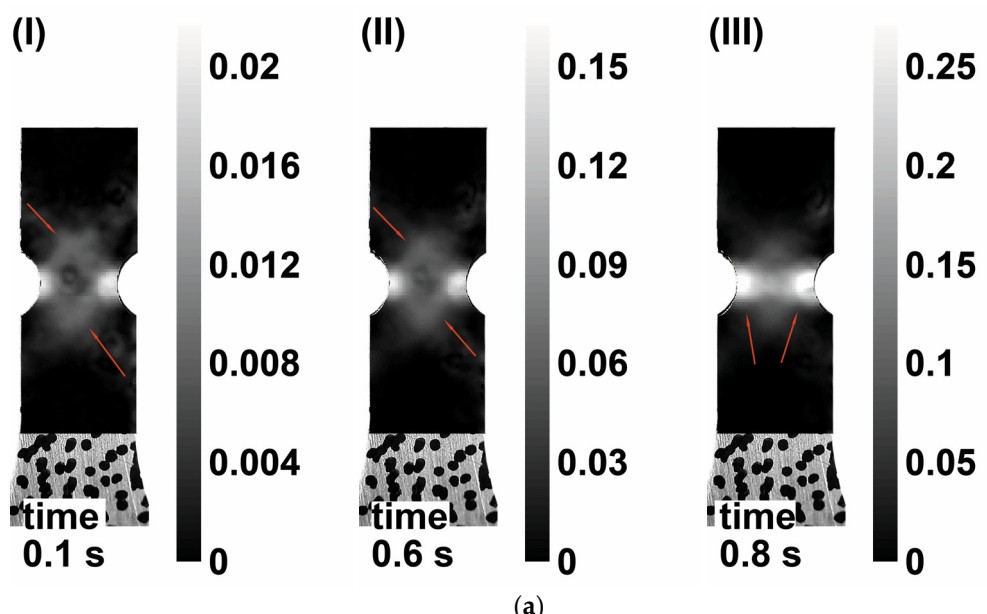

(a)

**Figure 8.** *Cont.*

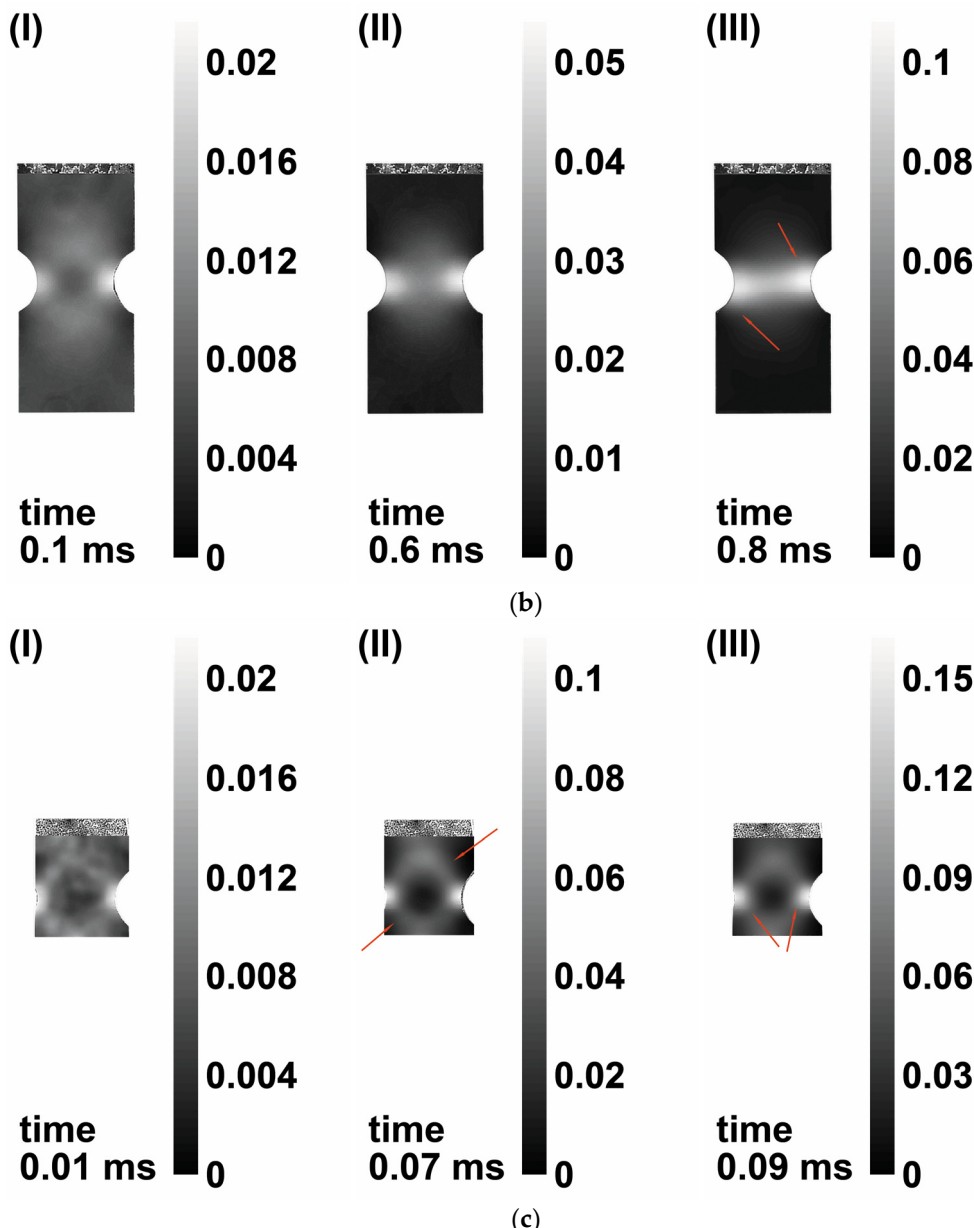

**Figure 8.** Fields of equivalent strain of specimens with the notch radius of 2.5 mm at (**a**) 0.1 s$^{-1}$, (**b**) 100 s$^{-1}$, and (**c**) 1000 s$^{-1}$.

Obtained data indicate that the development of plastic strain and the position of the forming crack depend on the stress triaxiality parameter and can vary significantly. Obtained data on the strain field evolution can be used to calibrate elastoplastic and damage models. Lindner et al. have shown the effectiveness in using DIC measurements to calibrate the mechanical response model of the titanium alloys. Our results are in agreement with those of Lindner et al. [20] for notched samples of Ti-6AL-4V alloy. A comparison of the data shown in Figures 4–8 indicate that the average strain to failure in notched samples does not provide sufficient information on the fracture behavior.

Our data on the influence of shear bands orientation on the separation crack position in Ti-5Al-2.5Sn during high speed tension are in agreement with results of Wang et al. obtained at dynamic compression [34]. Note that different isomechanical groups with HCP, BCC, and FCC crystalline lattice structure exhibit similar behavior, where the damage nucleates and the crack forms in the zone of shear band intersection under tensile loading conditions [35–52]. Rodriguez-Millan et al. [53] confirmed the complex effect of strain rate and stress triaxiality on the mechanical response of aluminum alloys

using experimental studies. Results of Bobbili et al. [51], Verleysen [52], and Zheng [54] confirm our results on the mutual influence of damage initiation processes and the plastic strain localization at the mesoscopic level for alpha, alpha + beta, and beta titanium alloys.

### 3.3. Fracture Surface Observation

An analysis of the DIC patterns clearly points to the crack nucleation position in the zone of intersection of the plastic strain localization bands. Note that crack formation depends on the stress state in the fracture zone. Moreover, the energy required for the fracture depends on the structure of the alloys and the physical mechanisms of ductile fracture [4,5,12].

Figure 9 shows configurations of formed macroscopic cracks crossing the gauge area of samples. The crack position and the fractal character of the fracture surface depend on the value of the stress triaxiality parameter and can vary significantly depending on the strain rate. Figure 9a–c demonstrates a possibility of transformations in spatial orientation of the fracture surface with the variation of stress triaxiality. Results on the alpha titanium alloy show that the integral area of the fracture surface depends not only on the stress triaxiality but also on strain rate. Thus, the energy required for the fracture surface formation depends also on both stress triaxiality and strain rate under investigated loading conditions.

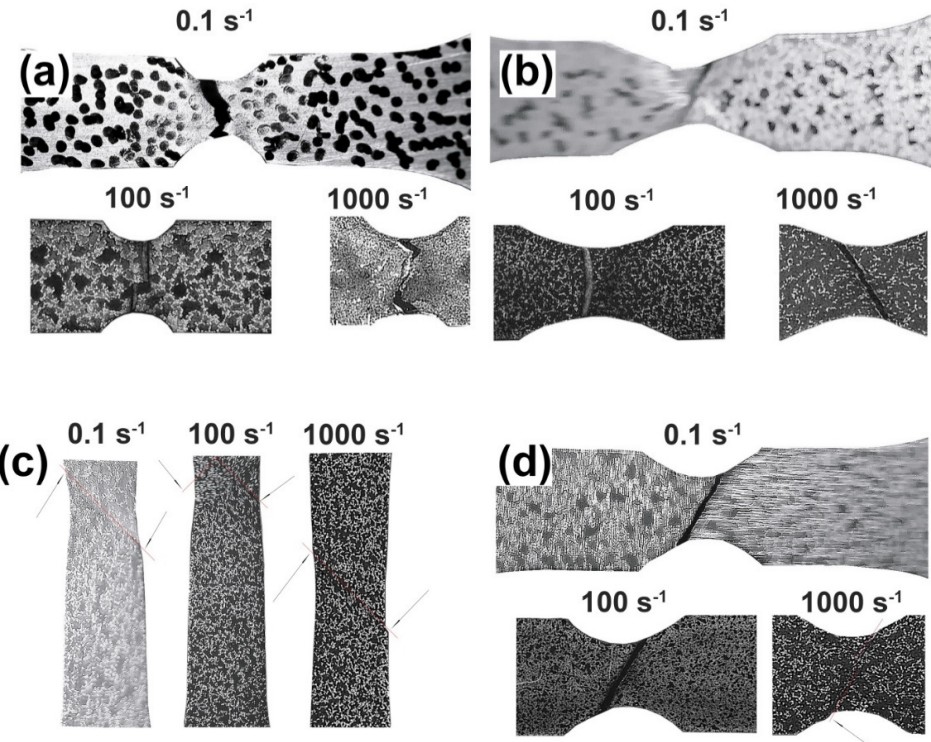

**Figure 9.** Frame of specimens before fracture; (**a**) notch radius $R = 2.5$ mm, (**b**) $R = 10$ mm, (**c**) specimens without notches, and (**d**) $R = 5$ mm.

The Keyence VHX-600D digital microscope (Keyence Corporation, Osaka, Japan) was used to observe the surface topography and quantify the roughness of the fracture surface.

Figure 10 shows fracture surface roughness profiles of Ti-5Al-2.5Sn notched and smooth samples after tensile tests at strain rates of 0.1, 100, and 1000 s$^{-1}$.

Methodologies to use the fracture surface roughness data to estimate GTN damage model parameters were discussed by Osovski et al. [35], Ponson et al. [36], and Needleman et al. [37]. Note that the analysis of fracture surface roughness parameters enables us to estimate the fractal dimension associated with the crack resistance of the material [38–43]. Surface profiles in Figure 10 indicate that

an increase in the stress triaxiality parameter from 0.33 to 0.55 leads to an increase in the roughness parameter $S_a$.

The roughness parameter $S_a$ was determined from the 3D images of the fracture surface by the relation $S_a = \frac{1}{MN} \sum_{i=1}^{N} \sum_{j=1}^{M} |h(x_i, z_j)|$.

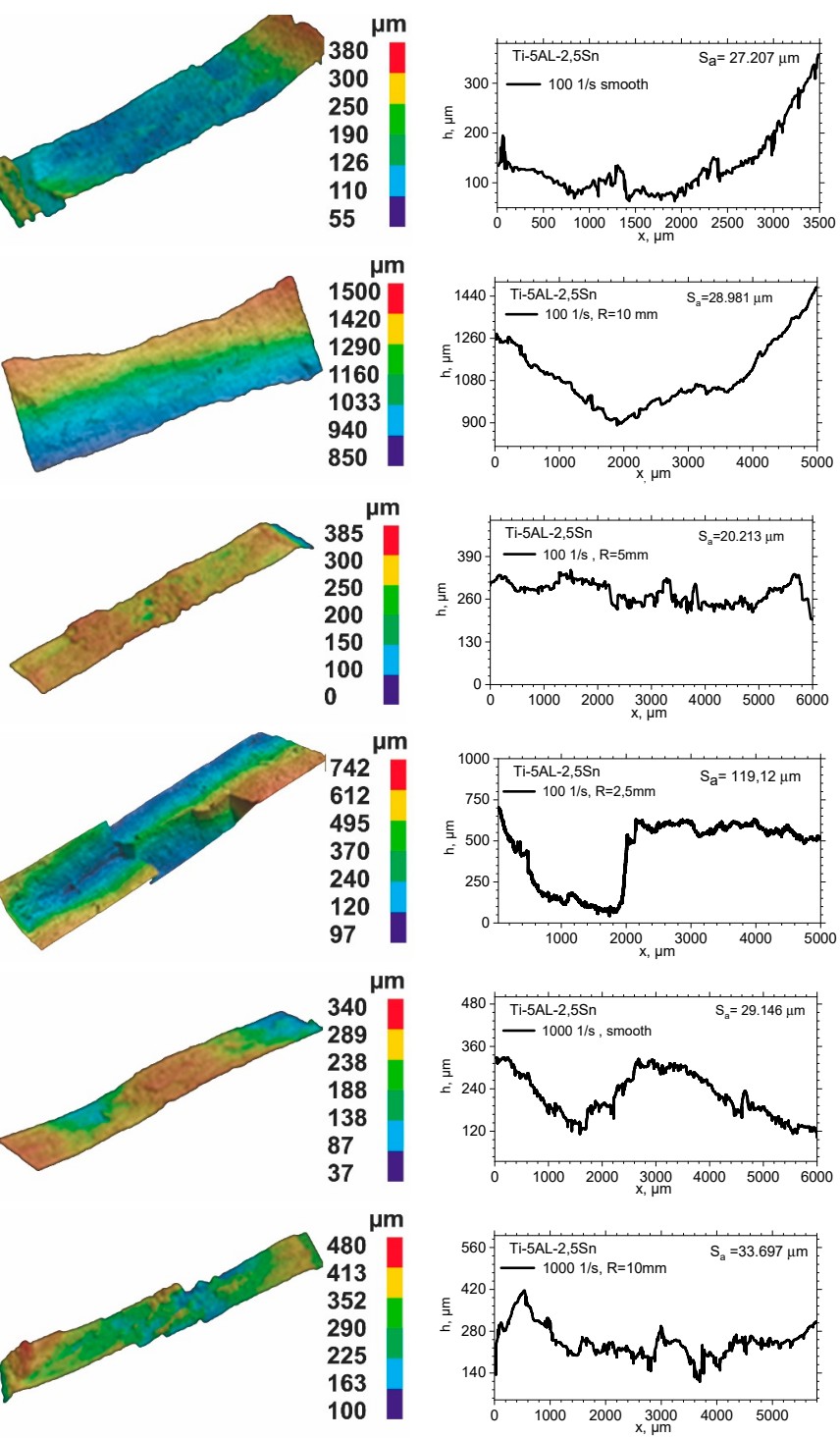

**Figure 10.** *Cont.*

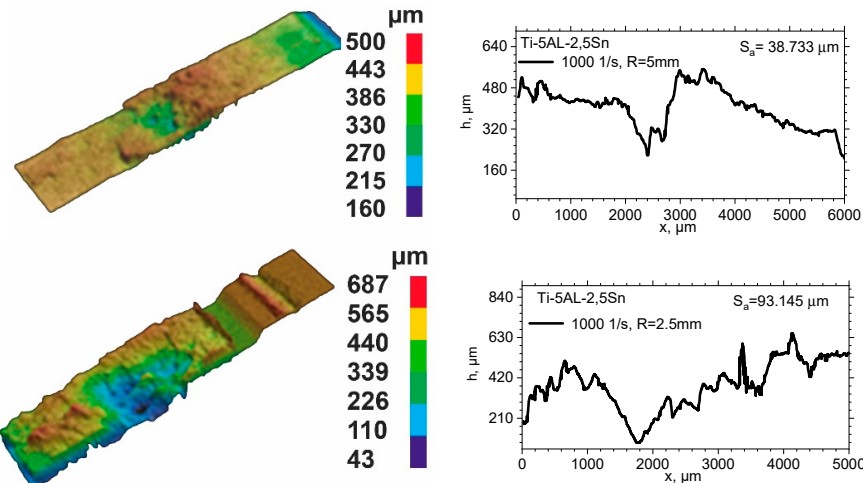

**Figure 10.** Roughness profiles of Ti-5Al-2.5Sn fracture surfaces. (**Left column**) shows 3D images of the fracture surface profile; (**right column**) shows profiles of height h along a straight line segment parallel to the axis OX and equidistant from the surfaces of the sample along a line equidistant from the sample surfaces.

Fractal dimension of the fracture surface enables us to evaluate the role of mechanisms of ductile and brittle fracture [37–41]. Fractal dimensions are used to describe the degree of tortuosity on boundaries of the fracture surface.

$$D_f = 1 - \frac{\log(\lambda/\lambda_0)}{\log s} \qquad (18)$$

where $D_f$ is the fractal dimension of the fracture surface, $\lambda$ is the measured length, $\lambda_0$ is a constant (Euclidean length), and $s$ is the measurement scale [42].

The fractal dimension $D_f$ increased linearly with increasing $S_a$, which could be described by the following Equation [38]:

$$D_f = D_0 + k_1 S_a, \qquad (19)$$

where $S_a$ is the surface roughness and $D_0$, and $k_1$ are coefficients.

The values of coefficients $D_0 = 2.0$ and $k_1 = 0.025$ 1/μm were estimated from the averaged data on the fracture surface roughness at strain rate of 100 s$^{-1}$. The values of $k_1$ are slightly less than obtained by Chang [27] for the titanium–iron alloy. Increasing fractal dimension indicates the intensification of void growth and tendency to ductile fracture. Liang et al. have shown a linear relationship between the strain to fracture and the fractal dimension of the fracture surface for magnesium alloys, which belong to the same isomechanical group as Ti-5Al-2.5Sn [40]. Results shown in Figures 3 and 10 for an alpha titanium alloy are consistent with this observation.

Figure 11 shows that the maximum and average values of the fracture surface roughness parameter $S_a$ increases with increasing stress triaxiality parameter $\eta$. This indicates that stress triaxiality affect not only void growth rate but also void nucleation. Void nucleation occurs in shear band intersections at low and intermediate homologous temperatures. Experimental results shown in Figure 11 indicate the nonmonotonous behavior of strain to fracture with increasing strain rate. This phenomenon can be related to more intensive temperature rise in shear bands, which contributes to thermally activated glide of dislocations over the obstacles and increased plasticity.

It should be noted that discussed observations are characteristic of coarse-grained alloys with an HCP lattice—FCC alloys behave in different manner [43].

Mean values of $S_a$ are given in Figure 10. Mean value of $S_a$ is reduced with increasing strain rate from 100 to 1000 s$^{-1}$.

Mean value of $S_a$ of the fracture surface of specimens with the notch radius, $R = 2.5$ mm, ($\eta \sim 0.55$) decreases by 1.8 times when the strain rate increases from 100 to 1000 s$^{-1}$.

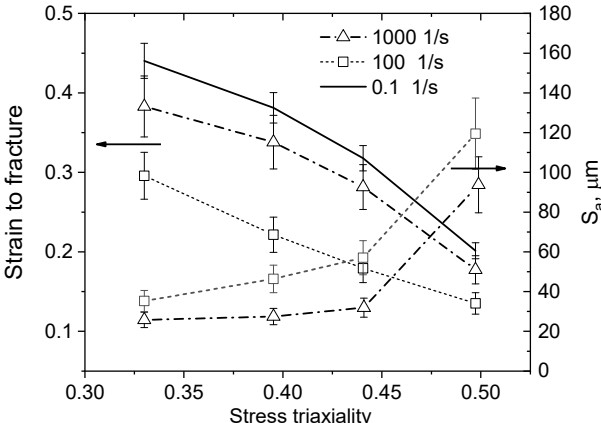

**Figure 11.** Strain to fracture versus stress triaxiality; fracture surface roughness $S_a$ versus stress triaxiality.

Obtained data are in agreement with those of Alonso et al. [44] who investigated the fracture of titanium alloys during drilling and cutting. Cutting and drilling of titanium is one of the most complex processes in the manufacture of structures for various purposes, including aircraft and medicine. Accounting for the stress state triaxiality is essential to improve cutting and drilling technologies of titanium alloys because of the complex stress state in the chip formation zone [44,45].

*3.4. Results of Simulation*

Numerical simulations of titanium alloy specimens subjected to tension were carried out to study damage kinetics and the influence of damage on mechanical behavior. Figure 12a shows the comparison between flow model predictions and experimental data of this work, as well as high temperature data, available in the literature for Ti-5Al-2.5Sn alloy by Doner et al. [46] and Zhang et al. [31,32].

Equation (13) describes the strain hardening, strain rate sensitivity, and the temperature sensitivity of the Ti-5Al-2.5Sn alloy within the condensed phase of undamaged material.

The model, with Equations (5) to (17), was used in numerical simulations of samples subjected to tension. The mesh model of notched specimen is shown in Figure 12b. The grid step size at the notch was set at 0.06 mm to ensure the convergence of obtained numerical results. Time step was determined over each cell using Courant–Friedrichs–Lewy condition. Fragmentation of specimens was simulated using the erosion technique. In all simulations performed, the internal energy of eroded elements did not exceed 5% of the internal energy of the computational domain.

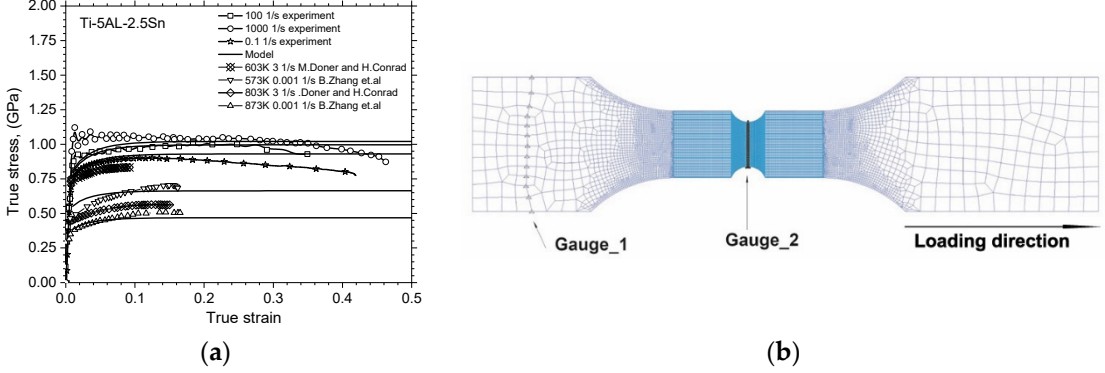

(**a**)  (**b**)

**Figure 12.** (**a**) Calculated stress versus strain and experimental true stress vs true strain curves of Ti-5Al-2.5Sn and (**b**) mesh model of specimen with notch.

Obtained results on the strain hardening behavior and temperature are in agreement with available data for CP Ti by Chichili [47], Roth [48], Salem [49], and Li [50]. Two different cross-sectional areas

were set to measure force values as shown in Figure 12b. Figure 13 shows the comparison between experimental data and calculated force–displacement diagrams.

The development of plastic flow in the gauge part of the specimen at high strain rates is accompanied by relaxation of shear stresses. As a result, release waves are generated. The propagation and interaction of these waves in loaded specimens result in force oscillations at the tail end of the specimen, which is in the elastic state. The force oscillations in the zone of the virtual gauge 1 were measured in the numerical simulation at tensile velocity of 20 m/s. Experimental curves of the measured average tensile forces versus displacement at a strain rate of 1000 s$^{-1}$ have a characteristic "hump".

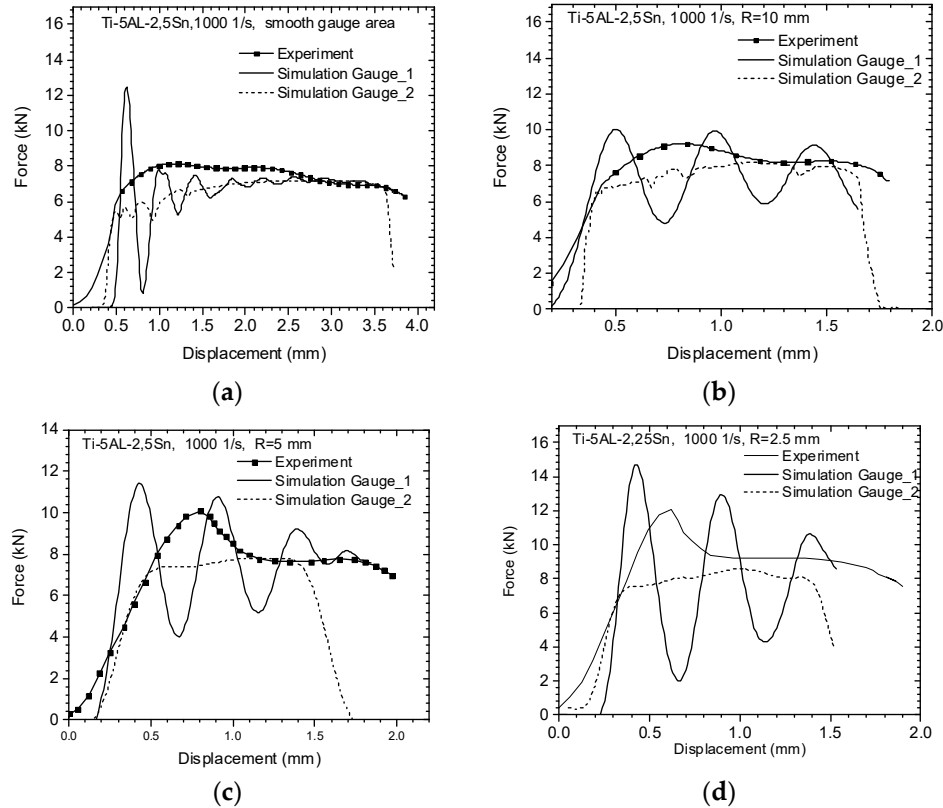

**Figure 13.** Calculated and experimental force–displacement curves; (**a**) specimen without a notch, (**b**) notch radius $R = 10$ mm, (**c**) notch radius $R = 5$ mm, and (**d**) notch radius $R = 2.5$ mm.

Figure 14 shows calculated values of the effective plastic strain obtained in the simulation. There was no obvious visual difference between simulated and observed configurations of the specimen. However, computed and measured strains were slightly different. The difference may be due to differences in time steps in the simulation and video registration.

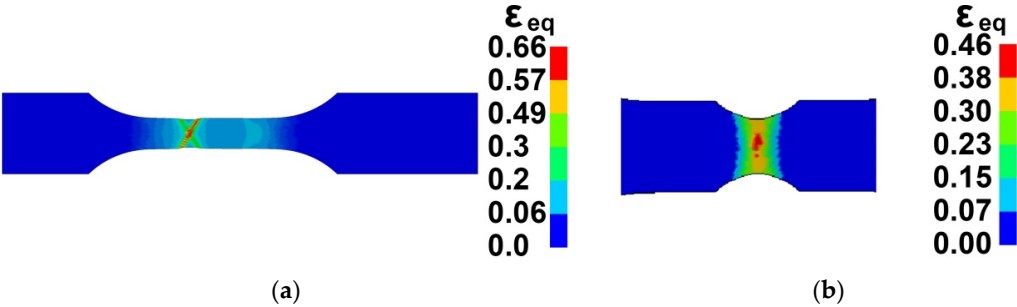

**Figure 14.** *Cont.*

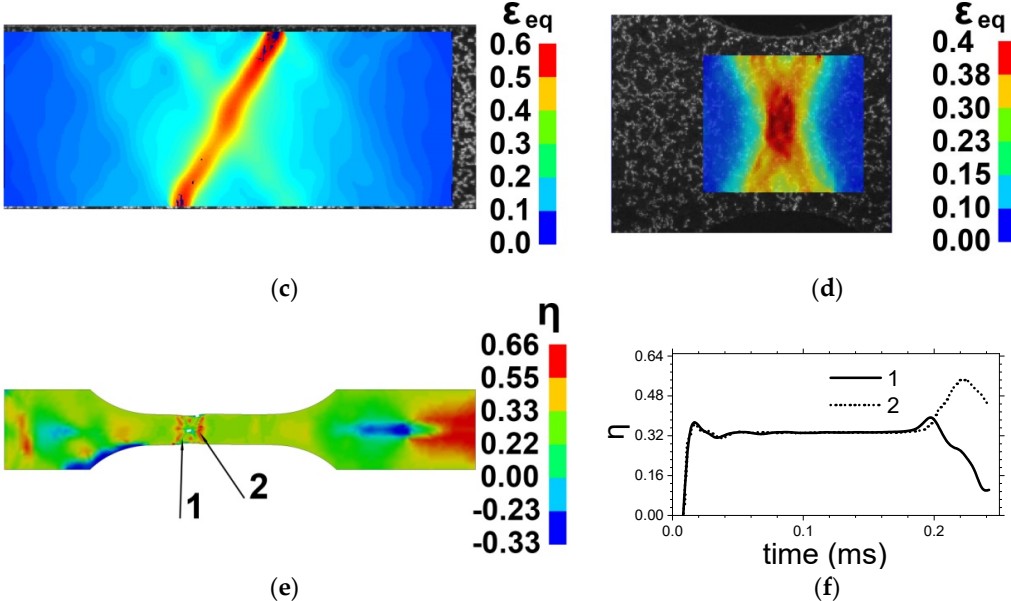

**Figure 14.** (**a**) Computed fields of the effective plastic strain for smooth specimens, (**b**) results for notched specimens with *R* = 5 mm; (**c**) experimental data obtained by digital image correlation (DIC) technique for smooth specimens of Ti-5Al-2.5Sn, (**d**) experimental data for specimens with the notch radius *R* = 5 mm, and (**e**) computed fields of the stress triaxiality factor. (**f**) Computed evolution of the stress triaxiality factor 1—in the fracture zone, 2—in the neck zone.

The notch radius and maximum values of the stress triaxiality parameter have a large effect on the strain distribution over the specimen gauge part.

Figure 15a shows the effective stress, damage parameter *f\**, and the effective plastic strain in the neck zone of Ti-5Al-2.5Sn specimen at strain rate 1000 s$^{-1}$. Two inclined stationary shear bands were formed in the neck zone. The simulations demonstrate the important role of strain localization phenomena in fracture processes. Figure 15b–d shows calculated parameters in the gauge area of specimens with notch radii of 10, 5, and 2.5 mm, respectively.

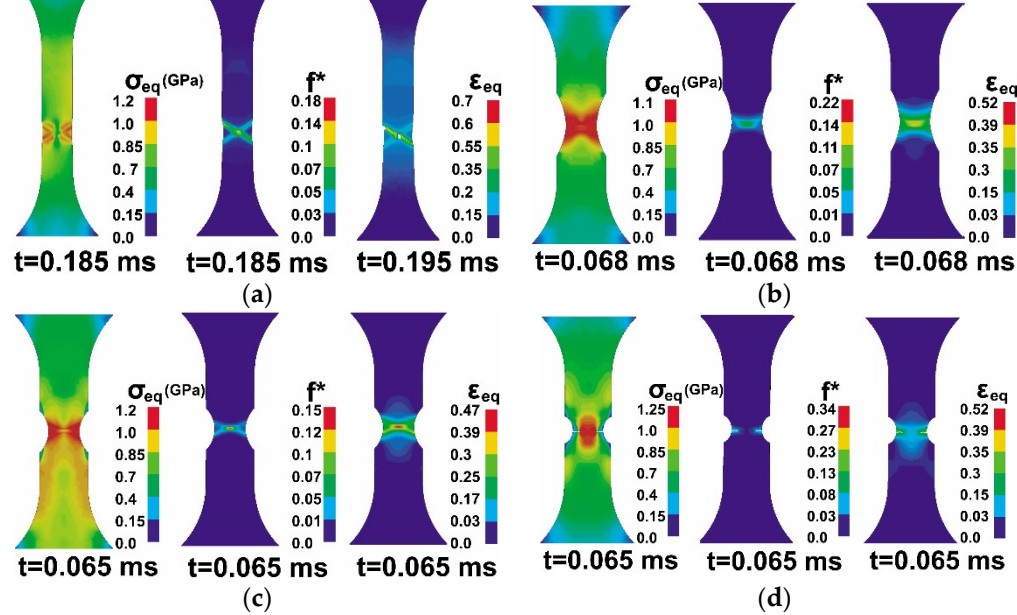

**Figure 15.** (**a**–**d**) Fields of calculated effective stress, damage parameter, and effective plastic strain in the specimen of Ti-5Al-2.5Sn under tension at 20 m/s.

The cracks position indicates a very strong correlation between the plastic strain localization and damage accumulation. The evolution of the calculated strain field and the fracture zone configuration are consistent with the data obtained by the DIC method at strain rates from 0.1 to 1000 s$^{-1}$ for notched specimens of various radii. Results indicate general laws governing the influence of the stress triaxiality factor $\eta$ on the ultimate mechanical characteristic of titanium alloys. Results are in qualitative agreement for the alpha + beta titanium alloy Ti-6Al-4V noted by Valoppi et al. [14] and near beta titanium alloys Ti-10-2-3 by Bobbili et al. [51].

The effective strain to failure of titanium alloys at room temperature decreases by 3.7 times with increasing stress triaxiality from 0.3 to 0.6 at strain rates of 0.1−1000 s$^{-1}$. Figures 13 and 14 show the calculated configuration of cracks. Obtained results on the influence of stress triaxiality on the strain to fracture of the alpha titanium alloys agree with the data of Bobbili [16]. Results also agree with data of Verleysen et al. [52] who showed that the phenomena of strain localization play a major role in the fracture process of Ti-6Al-4V titanium alloy at lower stress triaxiality. Figure 14f shows that the stress state triaxiality changes in the necking zone under tension. The variation of $\eta$ in the necking zone is caused by the shear stress relaxation at the formation of quasi-stationary shear bands.

## 4. Summary and Conclusions

In this paper, mechanical behavior of Ti-5Al-2.5Sn titanium alloy was studied under tension ($\eta$ = 0.33) and high triaxiality tension (0.33 < $\eta$ < 0.6) at practical valued strain rate range (0.1, 100, and 1000 s$^{-1}$) and at room temperature. Tensile tests were carried out on flat specimens using an Instron VHS 40/50-20 servo-hydraulic test machine.

Coherent series of strain fields up to the fracture were obtained and studied by means of Phantom 711 high-speed video-camera and Ncorr 2D DIC method [55].

The fracture surface relief of smooth and notched specimens was examined using the digital microscope Keyence VHX-600D (Keyence Corporation, Osaka, Japan).

The finite element simulation was carried out to analyze the processes of elastoplastic deformation and damage evolution in the alpha titanium alloy at strain rates of 0.1, 100, and 1000 s$^{-1}$. A modified elastoplastic model of damaged medium was used to describe the mechanical behavior of alpha Ti-5Al-2.5Sn titanium alloy.

Results of numerical simulation are in good agreement with the DIC data on the evolution of plastic strain fields and geometry parameters of the fractured zone in notched and smooth specimens during high-speed deformation.

Calculated fields of stresses, strains, and the stress triaxiality factor $\eta$ in the fracture zone complement obtained experimental information on the fracture of titanium alloys at high strain rates and stress triaxiality.

The evolution of strain fields at investigated loading condition indicates that large plastic deformation occurs in localization bands. The presence of well-defined strain localization bands is evidence of intensive strain hardening in alpha-titanium alloy occurring at mesoscopic level. This phenomenon results in significant heterogeneity in stress and strain fields.

Therefore, the fracture surface has a complex relief not only due to the mesh discretization of the material volume in the simulation. Obtained experimental data indicate the dependence of the fracture surface roughness parameter $S_a$ on the stress triaxiality $\eta$ in the titanium alloy.

Constitutive and failure model parameters can be determined on the base of tensile test results. The constitutive and fracture models have been validated by numerical simulation of tensile tests.

The experimental and numerical results obtained in this paper could be summarized as:

(1) Obtained experimental data indicate the dependence of the fracture surface roughness parameter $S_a$ on the stress triaxiality $\eta$ in the titanium alloy.

(2) Results confirm that the fracture of near alpha titanium alloys has ductile behavior at strain rates from 0.1 to 1000 s$^{-1}$, stress triaxiality parameter 0.33 < $\eta$ < 0.6, and temperature close to

295 K. The ductile to brittle transition has not been observed in Ti-5Al-2.5Sn titanium alloy under investigated loading conditions.

(3)  The alloy undergoes fracture governing by damage nucleation, growth, and coalescence in the localized plastic strain bands oriented along the maximum shear stresses. In the neck zone, which has formed upon tension of smooth specimens, the orientation angle of cracks and localized shear bands is close to 45° to the tensile axis.

(4)  Formation of adiabatic shear bands has not been detected in Ti-5Al-2.5Sn alloy under the studied loading condition. This has been confirmed by a rather high roughness of fracture surfaces. Trajectory of plastic strain localization bands and their intersection in separation zone determines the orientation of separation cracks near the stress concentrator zone.

(5)  Results of numerical simulation show that evolution of stresses and stress triaxiality $\eta$ near shear band intersection zone influence the damage accumulation and formation of the fracture zone trajectory.

Results obtained in this work can be useful for the modernization of tools and technologies of cutting and drilling of titanium alloys. The computational model can be used to analyze the mechanical behavior of the material and structures produced by selective laser melting of wire and Ti-5Al-2.5Sn titanium alloy powders.

**Author Contributions:** Conceptualization, V.V.S and V.A.S.; methodology, V.V.S., V.A.S. and E.G.S.; software, V.V.S. and V.A.S.; validation, V.V.S., V.A.S. and E.G.S.; formal analysis, V.V.S.; investigation, V.V.S., V.A.S., E.G.S.; resources, V.A.S.; data curation, V.V.S. and E.G.S.; writing—original draft preparation, V.V.S. and V.A.S.; writing—review and editing, V.V.S., V.A.S. and E.G.S.; visualization, V.V.S. and E.G.S.; supervision, V.A.S.; project administration, V.V.S.; funding acquisition, V.V.S. All authors have read and agreed to the published version of the manuscript.

**Funding:** This research was funded by the Russian Science Foundation (RSF), grant No. 16-19-10264.

**Acknowledgments:** The authors are grateful for the support of this research. Authors thank A. A. Kozulin, J. Starcevich, and A. V. Chupashev for the help in experimental tests.

**Conflicts of Interest:** The authors declare no conflict of interest.

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
