# Peer review of "Fracture of Titanium Alloys at High Strain Rates and under Stress Triaxiality"

_metals, doi:10.3390/met10030305_

Round 1
Reviewer 1 Report
Journal: Metals
Title: Fracture of Titanium Alloys at High Strain Rates and Stress Triaxiality
Authors: Vladimir V. Skripnyak, Evgeniya G. Skripnyak, VladimirA. Skripnyak
The reviewed paper deals with the effect of stress triaxiality on the mechanical behaviour and fracture mechanism of Ti-5Al-2.5Sn alloy, operating over a wide strain rate range. Tensile tests have been carried out on both smooth and notched specimens, recording videos by a high speed camera and utilizing the tool of Digital Image Correlation (DIC), to monitor the strain field upon plasticization in the neighbourhood of the notch. Analytical and numerical models have been developed as well, in order to get a better awareness of plasticization and fracture mechanisms, as effects of triaxiality and strain rate. Numerical model results have also been used, to calibrate the analytical ones. The combined outcomes of theoretical and experimental studies indicate that large plastic deformation occurs and is related to the generation of localization bands. This phenomenon has, in turn, a string correlation with the layout of the final fracture and is dependent on both triaxiality and strain rate.
The topic is surely consistent with the Aims and Scopes of Metals and the paper is generally well written, although the use of English must be improved, as some typos and errors are present throughout the text. Graphic quality is also rather good, but some pictures need improvements for the sake of Manuscript readability. In my opinion, the paper cannot be accepted in its current form and some major revisions must be required to the Authors. Recommendations for improvement are listed in the points below.
Title: in my opinion, it should more conveniently be turned into “Fracture of Titanium Alloys at High Strain Rates and under Stress Triaxiality”
Page 2 of 20, lines 66-67: “The tension tests were carried out in a range of strain rates (0.1 ̶ 1000 s-1) at room temperature using the Instron high speed testing machine VHS 40/50-20 with a 50 kN load cell.”
Have the tests been performed in the displacement controlled mode? If so, as I think, please also provide the controlled actuator velocity. Regarding the experimental tests, an important point is replication: how many replicated trials (per sample type and strain rate) have been run? This point must be highlighted in the paper.
Page 3 of 20, line 108: “xi is E is the specific internal energy”.
I think some words are missing here. Please, make sure that all the symbol are defined in the text. Perhaps, also adding a Nomenclature Section may help the reader.
Page 4 of 20, line 138: “Despite the fact that…” to be turned into “Although…”. The entire text must be checked for English.
Page 5 of 20: “Numerical values of model parameters are given in Table 2”.
Regarding the conducted numerical simulation, how many elements have been utilized? Has their dimension been refined in the neighbourhood of the notch or, generally, at sample gage?
Pages 6-7-8 of 20, Figure 3 and following ones. Do these plots refer to specific tests? Again, how many replicated trials have been conducted? Are the results referring to the same condition consistent one another?
Page 7 of 20, lines 203-205: “The elongation to fracture of the alloy under tension at strain rate from 0.1 to 100 s-1 decrease with increasing initial value of the stress triaxiality eta. Note that the indicated decrease in elongation to fracture has not monotonic behavior with increasing strain rate.”
Triaxiality is clearly related to sample geometry, with particular reference to that of the notch. The investigated triaxiality range has been provided above, but, in my opinion, this relationship and its related trend is still unclear and needs to be clarified by an additional graph. A further point arises from the assessment of the effects of the studied factors on the retrieved mechanical properties. These questions must be more properly addressed by the tools of statistical processing, in particular by Analysis of Variance (ANOVA) and subsequent significant tests. Therefore, the paper should be enriched by the outcomes of these analyses. Again, an important point is test replication, as the effect of the studied factor must be assessed as a comparison to the properly estimated experimental uncertainty.
Page 8 of 20, lines 227-228: “True strain – true stress diagrams of Ti-5Al-2.5Sn change when stress triaxiality parameter eta changes.”
Again, the relationship with triaxiality must be better clarified.
Page 8 of 20, lines 245-250: “Figure 5 (a), 6 (a), 7 (a), 8 (a) shows a series of three patterns of equivalent strain fields for the smooth specimen: onset the yielding, onset the neck formation, and prior the macroscopic fracture at 0.1 s-1. The first pattern indicates strain field heterogeneity to form in the work area. The second pattern shows the formation of conjugate localized shear bands. The third pattern shows the strain localization zone in which the fracture crack is formed.”
The cited figures must be redrawn, as they are indeed important for the full comprehension of the performed study, but they are currently barely visible. My suggestion is to use colours instead of black and white contrast, to provide larger images and to add arrows or circles (or similar sketches), to better highlight the most relevant patterns, which are mentioned in this passage (for instance, the shear bands). Moreover, the meaning of (a), (b) and (c) subfigures must be included in the captions. Please, also note that the passage “onset the yielding, onset the neck formation, and prior the macroscopic fracture at 100 s-1 Figures 5 (c), 6 (c), 7 (c), 8 (c) correspond to 1000 s-1 loading condition” is completely unclear.
Page 12 of 12, Figure 10: some strange symbols are present as title axes in the graphs: of course, they need to be amended.
Page 12 of 20, line 334: “The fractal dimension Df increased linearly with increasing Sa”
The meaning of “fractal dimension” must be clarified, for instance by a suitable sketch.
Page 13 of 20, lines 345-346: “Results indicate that maximum and average values of the fracture surface roughness parameter Sa increases with increasing stress triaxiality parameter eta.”
Again, this relationship vs. eta must be clarified, for instance by a graph.
Page 16 of 20, lines 402-403: “Simulation results show that the stress state triaxiality changes in the necking zone under tension.”
This result must be highlighted, adding arrows or circles to the pictures.
Page 16 of 20: Conclusions Section: try to summarize it, collecting the most relevant and impressive results of the conducted study in no more than 5 short bullet points.
Author Response
Dear Reviewer,
Thank you for your comments and suggestions. All of them were helpful for improving the manuscript. Please, find the responses in the attachment.
Sincerely yours,
Vladimir V. Skripnyak

Reviewer 2 Report
The authors investigate the effect of strain rate and stress triaxiality on the fracture behavior of titanium alloy by tensile testing of notched specimen. They also determined the model parameters for finite element analysis by comparing the strain distribution obtained by DIC method and load-displacement curves with calculated results. The information would be of interest and useful to the readers. I have only minor questions and comments listed below:
Line 108, “xi is E is …”, the xi should be explained. Figure 2, the axes (x, y and z) should be presented to understand the meaning of Eq. (17). Regarding DIC analysis, the space resolution of the pictures and subset size used in the analyses should be explained. Line 313 to 320 and Figure 10, which roughness parameter they used should be clearly presented. I guess the parameter is arithmetical mean height. If so, I guess what the right column in Figure 10 shows is not the roughness profile but the primary profile (or just “surface profile”) of the specimen. I think roughness profile (high-pass filtered surface profile) should be presented in the right column in Figure 10 if the authors want to compare arithmetical mean height. Figure 12, don’t the authors have any comments on the comparison between calculated and measured force-displacement curves? Why is the calculated force vibrating? Was the time step (or strain inclement) too large? Figure 13, the computed effective strain is slightly larger than measured strain. In the DIC analysis, the strain in thickness direction cannot be measured if the authors use 2D-DIC. How large strain is predicted in thickness direction in computed results? Figure 14, can the damage parameter predict the fracture point of the specimen?Author Response
Dear Reviewer,
Thank you for your comments and suggestions. All of them were helpful for improving the manuscript. Please, find the responses in the attachment.
Sincerely yours,
Vladimir V. Skripnyak

Reviewer 3 Report
The manuscript investigates the effect of stress triaxiality on mechanical behavior and fracture and the results are in agreement with the DIC data. The manuscript could be considered for publication after minor revision, the quality of presentation could be improved in particular you should revise the text about the simulation data for a better easy-reading.
- Check the format of equations
- Fig 10: Check the axis labes
Author Response
Dear Reviewers,
Thank you for your comments and suggestions. All of them were helpful for improving the manuscript. Please, find the responses in the attachment.
Sincerely yours,
Vladimir V. Skripnyak

Round 2
Reviewer 1 Report
Journal: Metals
Title: Fracture of Titanium Alloys at High Strain Rates and under Stress Triaxiality
Authors: Vladimir V. Skripnyak, Evgeniya G. Skripnyak, VladimirA. Skripnyak
I have carefully checked both the revised Manuscript and the enclosed Response Letter to my previous remarks. The paper has been indeed improved, moreover, according to the Journal Guidelines for the Authors, a point-by-point response has been provided, where the performed changes are properly described with the required detail.
However, in my opinion, some problems still arise from the quality of Figures 5 to 8. In fact, with respect to the original version, on one hand their clarity and readability have certainly been enhanced, but their organization is stull unsuitable for a Journal publication. For instance, considering sub-Figure 5 (a), for 0.1s^-1 strain rate, a collection of small images is presented at the left side, then two enlarged views are provided, moving rightwards for increasing deformation time (tagged as I and II). However, the last enlarged view for the highest deformation time (III) is delivered in the following row. Then, on the same (second) row, sub-Figure 5 (b), for 100 s^-1 strain rate, starts, again with a collection of small images and then with the image I for the shortest deformation time only. Conversely, images II and III are included in the third row. A similar description could be repeated for the other sub-Figures. In my opinion, the collections of small images must be deleted, as they do not add any further information. Moreover, each row must be devoted to (a), (b) and (c) sub-Figures, each including images I, II and III (not to be reported in different subsequent lines).
Therefore, I recommend further minor revisions, to address this remark.
Author Response
Dear Reviewer,
authors thank you for the constructive remark.
Sincerely yours,
Vladimir V. Skripnyak
